# Curcumin Stereoisomer, Cis-Trans Curcumin, as a Novel Ligand to A_1_ and A_3_ Adenosine Receptors

**DOI:** 10.3390/ph16070917

**Published:** 2023-06-22

**Authors:** Luke J. Hamilton, Mahesh Pattabiraman, Haizhen A. Zhong, Michaela Walker, Hilary Vaughn, Surabhi Chandra

**Affiliations:** 1Department of Biology, University of Nebraska at Kearney, Kearney, NE 68849, USA; hamiltonlj@unk.edu (L.J.H.); mjwalker1998@gmail.com (M.W.); vaughnh@lopers.unk.edu (H.V.); 2Department of Chemistry, University of Nebraska at Kearney, Kearney, NE 68849, USA; 3Department of Chemistry, University of Nebraska at Omaha, Omaha, NE 68182, USA; hzhong@unomaha.edu

**Keywords:** adenosine receptors, curcumin, vanilloid compounds, pain, photochemistry, A_1_AR, A_3_AR

## Abstract

Adenosine receptors (ARs) are being explored to generate non-opioid pain therapeutics. Vanilloid compounds, curcumin, capsaicin, and vanillin possess antinociceptive properties through their interactions with the transient receptor potential channel family. However, their binding with adenosine receptors has not been well studied. The hypothesis in this study was that a vanilloid compound, cis-trans curcumin (CTCUR), binds to each of the two Gi-linked AR subtypes (A_1_AR and A_3_AR). CTCUR was synthesized from curcumin (CUR) using the cavitand-mediated photoisomerization technique. The cell lines transfected with the specific receptor (A_1_AR or A_3_AR) were treated with CTCUR or CUR and the binding was analyzed using competitive assays, confocal microscopy, and docking. The binding assays and molecular docking indicated that CTCUR had Ki values of 306 nM (A_1_AR) and 400 nM (A_3_AR). These values suggest that CTCUR is selective for Gi-linked ARs (A_1_AR or A_3_AR) over Gs-linked ARs (A_2A_AR or A_2B_AR), based on our previous published research. In addition, the docking showed that CTCUR binds to the toggle switch domain of ARs. Curcumin (CUR) did not exhibit binding at any of these receptors. In summary, CTCUR and other modifications of CUR can be developed as novel therapeutic ligands for the Gi-linked ARs (A_1_AR and A_3_AR) involved with pain and cancer.

## 1. Introduction

Vanilloid compounds (VCs), characterized by the 4-hydroxy-3-methoxybenzyl group, include natural compounds such as vanillin, capsaicin, gingerol, and ferulic acid [1,2]. They have been used in traditional medicine, primarily for their antinociceptive and anti-cancer effects. Vanillin (Figure 1A) shows an antinociceptive effect, as demonstrated in mice with acetic acid-induced pain [3]; capsaicin (Figure 1B) combats postherpetic neuralgia and neuropathic pain [4]; 6-gingerol (Figure 1C) has anti-cancer and antinociceptive properties [5,6]; and ferulic acid (Figure 1D) is protective in skin cancer and inhibits neuropathic pain [7]. In addition, a VC which has gained importance in recent years is incarvillateine (Figure 1E), which comes from the Chinese herb, *Incarvillea sinensis,* and can inhibit formalin-induced pain [8,9].

Chronic pain is a debilitating condition which affects roughly 30% of the global population [10]. Several cellular receptors and signaling pathways have been shown to be involved in the transmission of pain. Inflammatory pain is mediated through cyclooxygenase (COX) enzymes which have been the target for the commonly used over-the-counter pain medications [11]. At the peripheral level, the transient receptor potential (TRP) family plays a prominent role mediating pain sensations [12]. Among the TRP family proteins, TRPA1 detects painful mechanical stimuli, TRPV1 mediates the pain response to heat [13,14], and TRPM8 mediates the pain response to cold [12]. The most common class of pain receptors for the design of therapeutics have been the opioid receptors, which act in both the peripheral and central nervous systems [15]. Though opioid analgesics have been the most effective, the number of deaths related to opioid overdoses have increased worldwide and more than tripled in the USA from 1999 to 2017 (Centers for Disease Control and Prevention, National Center for Health Statistics 2021). There was a slight decline in 2018 and 2019, but the numbers have again risen in 2020, possibly due to COVID-19. In response, studies have been initiated to discover new analgesics with a lower risk of addiction.

Among the non-opioid receptor families, adenosine receptors have shown potential as a target for analgesics [16,17,18,19]. Adenosine receptors include four subtypes: A_1_AR, A_2A_AR, A_2B_AR, and A_3_AR, all of which are G-protein-coupled receptors. A_1_AR and A_3_AR are coupled to inhibitory G proteins (G_i_), which depress the cAMP levels, while A_2A_AR and A_2B_AR are coupled to stimulatory G proteins (G_s_), which elevate the cAMP levels [20]. An agonist for A_2A_AR, regadenoson (Lexiscan) is used for perfusion imaging of the heart. The other approved AR-targeted drug is a non-selective agonist, namely adenosine itself (Adenocard, Adenoscan), which is used for the treatment of supraventricular tachyarrhythmia [21]. In addition, the FDA approved istradefylline, an A_2A_AR antagonist, for the treatment of Parkinson’s disease [22]. Although very few AR ligands have been approved for clinical use, all the AR subtypes have been investigated for their potential as targets for analgesic drugs.

Amongst the four AR subtypes, A_1_AR is the most important for the antinociceptive action of endogenous adenosine [23] as well as the antinociceptive effect of morphine [24]. The antinociceptive effect occurs when A_1_AR is bound to agonists [25,26,27]. The principal barrier to the development of A_1_AR-targeted analgesics is the concern over its cardiac side effects [18]. On the contrary, the increased interest in A_3_AR agonists as antinociceptives has been due to their high efficacy without the concerning cardiovascular side effects that were observed in the A_1_AR and A_2A_AR ligands [28,29]. Some evidence suggests that pain relief from the A_3_AR activation results from a modification of the release of the inhibitory neurotransmitter γ-aminobutyric acid (GABA) [28]. Other evidence suggests that the mechanism is the A_3_AR modulation of serotonergic and adrenergic neurons at the interface between the spinal cord and the medulla oblongata [18]. In any case, one of the most useful traits of A_3_AR agonists is that they block the pain induced by chemotherapy drugs without hindering the anti-cancer effect [16]. A_3_AR agonists have been tested in clinical trials for hepatocellular carcinoma [30] and psoriasis [31] and have been demonstrated to be safe. Finally, experiments with mice in withdrawal from morphine showed that A_3_AR agonists can lessen the withdrawal symptoms, indicating that even if A_3_AR cannot replace opioids, they may be a suitable adjunct therapy to opioids [24].

The previous literature has shown that the natural VC incarvillateine [9] and its synthetic structural analog ferulic acid dimer [32] both exhibit binding at the ARs using in vivo chronic and acute pain models, respectively. Curcumin (CUR), another VC, is effective in the treatment of autoimmune inflammation [33] and postoperative pain [34]. Based on the evidence that ARs are responsible for the antinociceptive action of incarvillatine [9], we initially hypothesized that ARs might play a role in the antinociceptive action of CUR as well, since CUR is also a VC. It has been demonstrated that CUR exerts its antinociceptive effects, in part, by acting as a COX pathway inhibitor [35] and a TRPV1 antagonist [36]. The original purpose of our study was to determine whether ARs play a role in the antinociception of CUR, in addition to COX and TRPV1. Our preliminary assays indicated that CUR did not bind well to the ARs, indicating that the ARs did not play a role in its antinociceptive properties. However, the same preliminary assays indicated that a synthetic analog of CUR, namely (1E, 6Z, namely, *cis-trans)* curcumin (CTCUR, Figure 2D, EZ-keto form) did bind to the ARs, with an IC_50_ in the micromolar range. Since the current literature lacks information on the interaction of VCs with ARs at the cellular level, we proceeded to characterize the interaction of CTCUR with the ARs. Our hypothesis for this study was that CTCUR (Figure 2D), a stereoisomer of (1E,6E)-CUR (Figure 3), can interact with the G_i_-linked ARs, A_1_AR and/or A_3_AR, and induce a downstream signaling action. The previous research from our lab has shown that CTCUR binds to the Gs-linked ARs, A_2A_AR and A_2B_AR. However, the results from this study indicated that CTCUR binds to Gi-linked ARs with a ten-fold higher affinity compared to the Gs-linked subtypes. This creates an avenue to explore this compound further for pathological conditions, including pain and cancer. This result also informs drug development by indicating that future attempts to target ARs with VCs should prioritize A_1_AR and A_3_AR over A_2A_AR and A_2B_AR.

## 2. Results

### 2.1. Receptor Binding and Cell Survival with CTCUR or CUR at A_1_AR

The binding curve with CTCUR or CUR at A_1_AR showed a dose-dependent curve, with CTCUR outcompeting the fluorescent ligand at 10^−5^ and 10^−4^ M (Figure 4A). As a result, the K_i_ of CTCUR at A_1_AR was calculated to be 3.06 × 10^−7^ M. The cell survival was performed at 2 h to simulate the binding assay incubation. CTCUR showed no reduction in cell survival from 10^−9^ M through 10^−4^ M at A_1_AR (Figure 4B). On the other hand, the K_i_ of CUR (Figure 4C) at A_1_AR was not determined due to toxicity (Figure 4D). There was a moderate toxicity at 10^−5^ M (58% survival compared to the control, Figure 4D) and a significant reduction at 10^−4^ M (37% survival compared to the control, Figure 4D).

### 2.2. Receptor Binding and Cell Survival with CTCUR or CUR at A_3_AR

CTCUR nearly completely outcompeted the fluorescent ligand at the 10^−4^ M concentration. The binding assays indicated that the K_i_ of CTCUR at A_3_AR was 4.00 × 10^−7^ M (Figure 5A). In terms of cell survival, CTCUR did not affect the cell growth in the A_3_AR cell line through all the tested concentrations (Figure 5B). There was no appreciable binding of CUR to A_3_AR. Hence, the K_i_ was not determined (Figure 5C). CUR showed a moderate reduction (65%) in cell survival at the highest tested concentration (10^−4^ M) (Figure 5D).

### 2.3. Binding Assay with CTCUR at Adenosine Receptors using Fluorescence Microscopy

The confocal microscopy data corroborated the results from the binding analysis. Minimal to no autofluorescence was observed using the media or CTCUR controls in both the A_1_AR and A_3_AR-transfected cell lines (Figure 6 and Figure 7). A Mann–Whitney U test of the CMGVs indicated that CTCUR at 10^−6^ M significantly blocked the binding of CA200623 to the A_1_AR-transfected CHO cells (*p* < 0.0001, Figure 6). An unpaired *t*-test of the CMGVs indicated that CTCUR at 10^−6^ M significantly blocked the binding of CA200623 to the A_3_AR-transfected CHO cells (*p* < 0.0001, Figure 7).

### 2.4. Docking Analysis with CTCUR at Adenosine Receptors

The docking studies indicated that CTCUR bound to A_1_AR to the extracellular end of transmembrane helix 3, the extracellular linking region between helices four and five, and the extracellular end of helix six [37] (Figure 8A). The docking indicated a ΔG for CTCUR at A_1_AR of −10.22 kcal/mol, which implied a 13% stronger binding compared to our experimental ΔG value, which was −8.89 kcal/mol. The experimental ΔG value was derived from the K_i_ value reported above. For the A_3_AR, the docking studies indicated that CTCUR bound to the extracellular end of transmembrane helix three, the center and extracellular end of helix six, the extracellular linking region between helices six and seven, and the extracellular end of helix seven (Almerico et al., 2013) (Figure 8B). The docking indicated a ΔG for CTCUR at A_3_AR of −10.16 kcal/mol, which implied a 20% stronger binding compared to our experimental ΔG value, which was −8.73 kcal/mol (derived from the K_i_ obtained from the binding assays).

## 3. Discussion

The principal discovery of this study was that CTCUR, a VC, bound to both A_1_AR and A_3_AR with the inhibitory constants of 306 nM and 400 nM, respectively. The binding affinities of CTCUR were compared to those of some established AR ligands, as shown in Table 1. The affinity of CTCUR for A_1_AR was three-fold weaker than that of the adenosine. The similarity of affinity for A_3_AR between CTCUR and adenosine was even closer, with a difference less than two-fold. Furthermore, CTCUR had a substantially greater affinity for the ARs of all the subtypes when compared to caffeine and theophylline (Table 1). Thus, it was reasonable to hypothesize that CTCUR might have a biologically relevant effect, since its affinity for ARs was comparable to that of biologically active compounds. In contrast, it should be noted that the affinity of CTCUR for A_1_AR and A_3_AR was substantially less than that of the previously developed ligands, such as NECA, Tecadenoson, and IB-MECA, which have affinities in the nanomolar range (Table 1). The goal of this study was to elucidate the molecular-level interaction between an important class of medicinal compounds (VCs) and an important receptor family (ARs). The specific medicinal properties of CTCUR, if any, have not yet been tested.

In our previous study, we examined the interaction of CTCUR with the two G_s_-linked ARs [38]. The data from the binding assays, confocal microscopy, and docking all supported CTCUR as selective for the two G_i_-linked subtypes (A_1_AR and A_3_AR) over the two G_s_-linked subtypes (A_2A_AR and A_2B_AR) (Table 2). The cytotoxicity assays further indicated that CTCUR was less toxic compared to its parent compound, CUR (Figure 4C,D and Figure 5C,D). Thus, the binding assays outcompeted CTCUR in the fluorescent ligand but not toxicity.

Table 3 shows that other than the relative error (ΔΔG) for tecadenoson for the A_1_AR model, all the other errors (ΔΔG) between ΔGpred (binding score) and ΔGexp for the A_1_AR and A_3_AR subtypes were less than 2.0 kcal/mol, suggesting that the Glide Dock program can predict the binding affinity measured by ΔGpred (binding score), which was very close to the experimentally obtained data and suggested the reliability of the Glide Dock program.

The orthosteric site of A_1_AR consisted of eleven residues, which were Val87, Thr91, Phe171, Glu172, Met180, Trp247, Leu250, Asn254, Ile274, Thr277, His278 [43]. CTCUR was predicted to interact with seven of these eleven residues as well as other residues in the orthosteric site, with His278 and Phe171 providing the H-bond interactions and aromatic-aromatic interactions (Figure 8A), indicating that CTCUR occupies the orthosteric site when it binds to A_1_AR. Although no crystal structure of A_3_AR was reported, the docked pose of CTCUR at the A_3_AR binding pocket showed that CTCUR may also bind to the orthosteric binding sites consisting of Ser73, Thr94, Gln167, Met177, Trp243, Ser247, Asn250, and Phe168 (Figure 8B).

ARs are part of the rhodopsin-like family of GPCRs, and as such it is thought that the Trp residue within the C/S W X P domain of transmembrane helix six (that is, Trp247) is important for the A_1_AR activation [37,44,45]. Our previous study showed that Trp247 provided key interactions with A_2A_AR and A_2B_AR [38]. Our current docking study also identified Trp247 of A_1_AR as a binding residue to the phenol group of the CTCUR (Figure 8A). These data supported the hypothesis that CTCUR was an agonist of ARs. Although we did not see statistically significant changes in cAMP production in response to CTCUR, and therefore, cannot report an EC_50_ value (see Appendix A), the overall trend in our results suggested that CTCUR was a weak agonist of ARs. In this study of Gi-linked ARs, we saw decreases in the cAMP production in response to 10 μM of CTCUR. This pattern of changes in the cAMP concentration in the AR-transfected CHO cells treated with CTCUR was comparable to the pattern seen in a study of AR-transfected CHO cells treated with adenosine [46]. From the perspective of targeting the Gi-linked ARs for antinociception, a weak agonistic activity was sometimes better than a strong agonistic activity. The previous work with partial A_1_AR agonists has shown that they can affect analgesia without causing the cardiovascular side effects associated with full agonists [47]. Future research could focus on modifying the region of CTCUR that binds to transmembrane helix six to generate curcuminioids with an optimal receptor activation at the Gi-linked ARs.

Evidence suggests that the antinociceptive effects of A_3_AR agonists may be co-mediated by N-type voltage-gated Ca^2+^ channels [48]. In general, it is important to realize that several molecular signaling pathways can mediate antinociception, of which ARs are only one. For VCs specifically, their antinociceptive effects are sometimes mediated by TRPV1 [49]. However, the question of whether TRPV1 is the primary mechanism of action for VC antinociception remains to be answered [36]. A comparison between the binding of CTCUR to ARs and the binding of other VCs to TRPV1 is, therefore, warranted. For example, the interaction of capsaicin with TRPV1 had an EC_50_ of 712 nM [50] and the interaction of CUR with TRPV1 had an EC_50_ of 67 nM [36]. In contrast, the EC_50_ values of CTCUR at A_1_AR and A_3_AR were much higher, namely 5725 nM and 9315 nM, respectively. This suggests that, although ARs may play some role in the mechanism of antinociception for VCs, TRPV1 most likely plays a more important role. Future experiments should aim to quantify the binding affinity and receptor activation of CTCUR at TRPV1.

It is known that A_3_AR plays a role in pain regulation. The A_3_AR agonists MRS5980 and CI-IB-MECA significantly reduced 2,4-dinitrobenzenesulfonic acid-induced visceral pain in different phases of inflammation and the effect was completely abolished by the selective A_3_AR antagonist MRS1523 [48]. In addition, the anti-nociceptive effects of the A_1_AR agonist and the A_3_AR agonist can be explained by the effects of the AR ligands on TRPV1 [51]. To investigate the effects of the vanilloid compounds on A_1_AR and A_3_AR, we built and docked vanilloid compounds (vanillin, capsaicin, 6-gingerol, and ferulic acid) to A_1_AR and A_3_AR. Table 3 shows that the docking scores of capsaicin, 6-gingerol, and ferulic acid against A_3_AR and A_1_AR were predicted to be as potent as adenosine and tecadenoson, which were verified as the A_1_AR and A_3_AR agonists.

ARs have medicinal applications across several organ systems [52]. For example, A_1_AR agonism in the heart can suppress the atrioventricular node activity [53] and the sinoatrial node activity [54], and thereby correct tachycardia. In a Phase II clinical trial, the A_1_AR agonist trabodenoson was shown to decrease the intraocular pressure in patients with either ocular hypertension or primary open-angle glaucoma [55]. Additionally, in a study of murine kidneys, the A_1_AR agonist CCPA was found to protect against renal ischemia-reperfusion injury by inducing sphingosine kinase-1 [56].

Since A_3_AR is overexpressed in cancer cells, A_3_AR agonists are effective in combating cancer of the skin, prostate, colon, and liver [57]. A_3_AR agonists can also reduce inflammation, as shown by the success of the A_3_AR agonist piclidenoson in ameliorating rheumatoid arthritis [58]. Finally, in a rat model of cerebral ischemia, the A_3_AR agonist Cl-IB-MECA suppressed programmed cell death and reduced the severity of cerebral infarction [59]. VCs may, thus, act through ARs in other ailments besides pain.

## 4. Materials and Methods

### 4.1. Chemical Synthesis

CTCUR was synthesized using photochemical cavitand-mediated isomerization, as published by our group earlier [38]. The yellow-colored CTCUR was isolated in an 8% yield. The presence of CTCUR was confirmed using proton nuclear magnetic resonance (^1^H NMR) and gas chromatography–mass spectrometry (GCMS). The gas chromatography column was Agilent HP-5 (5%-phenyl)-methylpolysilane nonpolar (Agilent, Santa Clara, CA, USA). The initial temperature was 100 °C, the initial time was 1 min, the heating rate was 10.0 °C/min, the final temperature was 300 °C, and the final time was 5 min. The retention time of CTCUR was 6.8 min.

The ^1^H NMR (CDCl_3_, 400 MHz) data were as follows: 8.08 ppm (m, 1H); 7.98 ppm (d, 1H J = 15.3 Hz); 7.66 ppm (d, 1H,); 7.40 ppm (t, 1H); 7.11 (m, 2H); 7.05 (s, 1H); 6.94 ppm (d, 1H, J = 12.5 Hz); 6.48 ppm (d, 1H, J = 15.3 Hz); 6.32 ppm (d, 1H, J = 12.5 Hz); 3.98 (s, 3H); and 3.93 (s, 3H). The mass spectrometry data were as follows, in the format m/z (relative %, ion): 367 (5, M^+^), 351 (3, M^+^ − OH), 245 (5, M^+^ − Ar-OMe), 191 (100, M^+^ − CH=CH-CO), 177 (90, M^+^ − CH_2_), 149 (30, −CO), and 123 (10, M^+^ − CH=CH).

### 4.2. Cell Culture

The Chinese hamster ovary-K1 (CHO-K1) cell lines transfected with human A_1_AR or human A_3_AR, respectively, were purchased from Perkin Elmer (Waltham, MA, USA). The cells were cultured in Ham’s F12 medium supplemented with 10% FBS. For all the cell lines, the first two passages after thawing were cultured without antibiotics to allow for recovery from the thawing stress. Then, all the subsequent passages were cultured with 400 µg/mL of geneticin (Thermo Fisher Scientific, Waltham, MA, USA) to select the transfected cells. All the cell lines were cultured at 37 °C in a humidified atmosphere with 5% CO_2_. The cells were passaged and used for the experiments when they were 70–80% confluent, and a new vial of cells was recovered when the liquid nitrogen cells reached ~20 passages.

### 4.3. Competitive Binding Assays

The cells were seeded at 30,000 cells per well onto a white-walled, opaque-bottomed 96-well plate and were allowed to incubate for two days to reach a 90% to 100% confluence. The cells were then treated with either (A) media only, (B) media + test compound at 10^−5^ M, (C) media + fluorescent compound at 60 nM, or (D) media + fluorescent compound at 60 nM + test compound at varying concentrations. The test compound was either CUR or CTCUR, and the fluorescent compound was a CA200623 CellAura fluorescent adenosine agonist, a derivative of 5′-N-Ethylcarboxamidoadenosine (NECA) (HelloBio, Princeton, NJ, USA). After a 2 h incubation, the cells were washed once using 1X PBS. Dulbecco’s Modified Eagle Medium (DMEM) without a phenol red indicator was then added (100 μL per well) to prevent desiccation during reading. The plate was read using a Synergy H1 microplate reader (BioTek, Winooski, VT, USA) in a top read mode. This method of high throughput screening of the compounds via competition with the fluorescent ligands of ARs has been validated by previous researchers [60].

To generate a binding curve from the competitive binding assay (CBA) data, a percent fluorescence change was calculated for each treatment and plotted as a % change in fluorescence compared to no treatment control (Equations (1)–(3)). These values were, in turn, used to calculate the inhibitory constant (K_i_).
(1)$$\mathrm{Control}\;\mathrm{fluorescence}=\left(\mathrm{fluorescence}\;\mathrm{of}\;\mathrm{CA}200623\right)-\left(\mathrm{fluorescence}\;\mathrm{of}\;\mathrm{media}\right)$$
(2)$$\begin{array}{c} \mathrm{Treatment}\;\mathrm{fluorescence}\\ \;\;\;\;\;\;=\left(\mathrm{fluorescence}\;\mathrm{of}\;\mathrm{CTCUR}+\;\mathrm{CA}200623\right)\\ \;\;\;\;\;\;-\left(\mathrm{fluorescence}\;\mathrm{of}\;\mathrm{CTCUR}\right) \end{array}$$
(3)$$\%\;\mathrm{fluorescence}\;\mathrm{change}=\frac{\mathrm{Treatment}\;\mathrm{fluorescence}}{\mathrm{Control}\;\mathrm{fluorescence}}\times 100\%$$
(4)$$K_{i}=\frac{{IC}_{50}}{1+\frac{\left[L\right]}{K_{d}}}$$
where K_i_ is the inhibitory constant expressed in nM, *IC*_50_ is the test compound concentration when half of the binding is inhibited, [*L*] is the concentration of the fluorescent compound, and *K_d_* is the dissociation constant of the fluorescent ligand, equal to $\frac{{\left[\mathrm{fluorescent}\;\mathrm{ligand}\right]}^{x}{\left[\mathrm{receptor}\right]}^{y}}{\left[\mathrm{ligand}-\mathrm{receptor}\;\mathrm{complex}\right]}$.

### 4.4. Docking Studies

#### 4.4.1. Preparation of the Protein Structures

The X-ray crystal structure of A_1_AR (Protein Data Bank ID: 5UEN) (Glukhova et al., 2017) was retrieved from the RCSB Protein Data Bank. The covalent bond between the ligand (DU1) and Tyr271was disconnected to prepare the protein for the docking studies. The downloaded structure was prepared in the Molecular Operating Environment (MOE) (Chemical Computing Group Inc.; Montreal, QC, Canada, 2020), and the side chain and ligand atoms were optimized, first using the main chain atoms fixed in order to reduce steric repulsion and at the same time to minimize the perturbation to the main chain structure. After that, the whole structure was optimized using the Assisted Model Building with Energy Refinement (AMBER14:EHT) force field. There were no crystal structures of A_3_AR. Thus, a homology model for A_3_AR was built using the homology model module in the MOE and using the human A_3_AR sequence (GenBank: CAA54288.1) as the query sequence. The template structure to build the homology models was the A_1_AR crystal structure 5UEN. After the homology model (HM_A3) was built, the structural alignment to the template 5UEN was conducted in the MOE to evaluate the quality of the model proteins. The model protein was optimized first with a fixed main chain atom and then further optimized by relaxing all the atoms. HM_A3 was aligned to 5UEN, and the ligand in 5NM4 was adopted to the homology model proteins to help identify the binding pocket for the docking studies.

The MOE-optimized proteins (A_1_AR and A_3_AR) were imported to Maestro 12.4 (Schrödinger LLC; New York, NY, USA, Release 2020-2) and were subsequently prepared using the Protein Preparation Wizard in the Schrödinger software suite (Schrödinger LLC; New York, NY, USA, Release 2020-2). In the Protein Preparation Wizard, the side-chain structures of glutamine and asparagine were allowed to flip to maximize the H-bond interactions. The proteins were optimized using a default with the optimized potentials for liquid simulations (OPLS) force field in the MacroModel module in the Schrödinger software (Schrödinger LLC; New York, NY, USA, Release 2020-2).

#### 4.4.2. Preparation of the Ligand Structures

The ligands from the reported literature were built in the MOE to validate the docking method. The ligands used in this study were adenosine, tecadenoson, caffeine, theophylline, ZM241385, and istradefylline [61]. We also built the vanilloid compounds vanillin, capsaicin, 6-gingerol, and ferulic acid to study the effect of vanilloids on the ARs. To study the binding selectivity between the AR subtypes, compounds A, B, and KW3902 (NAX) were built as controls, along with CTCUR in its EZ-keto form (Figure 2). All the ligands were built in the MOE (Chemical Computing Group Inc.; Montreal, QC, Canada, 2020) and were optimized using a Merck Molecular Force Field 94 (MMFF94) with the default settings. The optimized ligands were then exported to Maestro for docking studies. All the imported ligands in Maestro were further minimized using the OPLS force field in the MacroModel module in the Schrödinger software (Schrödinger LLC; New York, NY, USA, Release 2020-2).

#### 4.4.3. Glide Docking Procedures

Two grid files for A1 and HM_A3 were generated using the Glide Grid Generation protocol (Schrödinger LLC; New York, NY, USA, Release 2020-2) with the bound ligands as the centroids. The ligands were docked to each of the grid files, which defined the binding pockets of the respective proteins. During the docking process, the scaling factor for the receptor van der Waals for the nonpolar atoms was set to 0.8 to allow for some flexibility of the receptor, and an extra precision method was used to calculate the docking scores. All the other parameters were set to default. The binding affinity was expressed in terms of the change in free energy (ΔG). The more negative the ΔG, the more favorable the interaction of the complex. To compare the ΔG values derived from docking with the *K_i_* values derived from the CBAs, the *K_i_* values were converted to ΔG values (Equation (5)).
(5)ΔG=RTlnKi kcalmol

### 4.5. Confocal Fluorescence Microscopy

The cells were seeded onto 12-well glass-bottom plates (Cellvis, Mountain View, CA, USA) at 375,000 cells per well. The cells were allowed to incubate for two days to reach a 90% to 100% confluence. On the day the images were collected, the cells were treated with either (A) media only, (B) CTCUR at 1 µM, (C) CA200623 at 60 nM, or (D) CA200623 at 60 nM + CTCUR at 1 µM. The microscopy protocol was a modification of a previously published protocol [62]. The cells were incubated for 2 h, washed once with 1X PBS, immersed in DMEM without a phenol red indicator, and viewed using a 60× oil immersion objective and an Olympus FV3000 laser scanning confocal microscope (Olympus, Tokyo, Japan). The images were acquired at a resolution of 640 by 640 pixels. At each location, a stack of 60 images at 0.5 μm intervals in the *z*-axis were acquired. All the images were processed using the CellSens Dimension Desktop V1.18 (Olympus, Tokyo, Japan). A constrained iterative deconvolution was performed on each stack. To reduce the bias in selecting individual images out of the stacks, we first examined all the stacks to see which image was, on average, the best in each stack. For the A_1_ cells, the average best image was 64% of the way through the stack. For the A_3_ cells, the average best image was 61% of the way through the stack. We selected the average best image from each stack and used only those images for the quantitative analysis.

The quantitative analysis was performed using ImageJ with a modification of a protocol from the Queensland Brain Institute. For each image, nine cells were selected for analysis. To reduce bias, the cells were selected according to a list of pixel coordinates that were determined before the images were analyzed. The background fluorescence level was determined by measuring the mean gray value (MGV). The corrected mean gray value (CMGV) for each cell was calculated according to Equations (6) and (7)

The CTCF is the corrected total cell fluorescence. The MGV is the mean gray value.
(6)$$\begin{array}{c} \mathrm{CTCF}=\left(\mathrm{Integrated}\;\mathrm{density}\right)-\left(\mathrm{Area}\;\mathrm{of}\;\mathrm{selected}\;\mathrm{cell}\right)\\ \;\;\;\;\;\;*\left(\mathrm{MGV}\;\mathrm{of}\;\mathrm{background}\;\mathrm{fluorescence}\right) \end{array}$$

The CMGV is the corrected mean gray value. The CTCF is the corrected total cell fluorescence.
(7)$$\mathrm{CMGV}=\frac{\mathrm{CTCF}}{\mathrm{Area}\;\mathrm{of}\;\mathrm{selected}\;\mathrm{cell}}$$

### 4.6. Cytotoxicity Assays

The cytotoxicity assays were performed using Prestoblue dye as per the manufacturer’s instructions. Since the goal of these cytotoxicity assays was to demonstrate that the binding observed in the CBAs was not due to cell death, the protocol was modified to mimic that of a CBA. The cells were seeded at 30,000 cells per well in 96-well plates, incubated until they were 70–80% confluent, and then treated using different concentrations of CUR or CTCUR for 2h. CUR and CTCUR were dissolved in DMSO at a concentration of 0.1 M and diluted for the assays as required. Cell survival was monitored using PrestoBlue dye (Thermo Fisher Scientific, Waltham, MA, USA) per the manufacturer’s instructions, and the fluorescence was measured at 560 nm/590 nm (excitation/emission) using a Synergy H1 microplate reader and the Gen5 software (BioTek, Winooski, VT, USA). The percent change was calculated relative to the average for the control cells (no treatment).

### 4.7. Statistical Analysis

For each CBA dataset, the inhibitory constant was calculated according to the Cheng–Prusoff equation (Stoddart et al., 2016) (Equation (4), see Section 2.3). For fluorescence microscopy, the CMGVs from the CA200623-only and the CA200623/CTCUR combination treatments were compared with the unpaired *t*-tests. If the assumptions of the t-test were violated, a Mann–Whitney test was used. For the cyclic AMP assays, the treatments were compared via an ordinary one-way ANOVA with a Tukey multiple comparisons test. If the assumptions of the ANOVA were violated, a Kruskal–Wallis test with a Dunn’s multiple comparisons test was used. The K_i_ calculations, the tests to compare the treatments, and the tests to assess the normality and homoscedasticity of the datasets were all performed using Prism 8.4.3 (GraphPad, San Diego, CA, USA). For all the analyses, *p* < 0.05 was considered significant (α = 0.05).

## 5. Conclusions

CTCUR binds with high affinity to the adenosine receptors A_1_AR and A_3_AR and has the potential to be developed as a novel non-purinergic adenosine receptor ligand. When combined with the previous work from our lab, the results of this study demonstrated that CTCUR was selective for A_1_AR and A_3_AR (the subtypes linked to Gi) over A_2A_AR and A_2B_AR (the subtypes linked to Gs). To our knowledge, this was the first time that the receptor binding and activation profile of any VC has been quantified in vitro for all four AR subtypes. Since VCs are widely known to be antinociceptive, and since ARs are widely known to mediate antinociception, the results of this study of the molecular-level interaction between VCs and ARs are useful to pharmacologists, and particularly to pharmacognosists, who study pain management. Future studies of the interaction between VCs and ARs should prioritize A_1_AR and A_3_AR over A_2A_AR and A_2B_AR.

## Figures and Tables

**Figure 1 pharmaceuticals-16-00917-f001:**
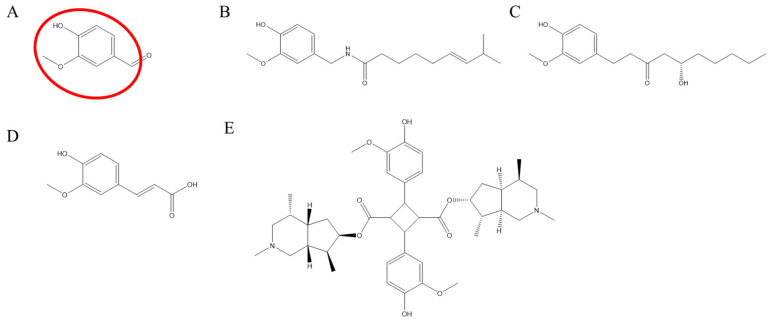
Chemical structures of (**A**) vanillin (PubChem CID 1183) with the 4-hydroxy-3-methoxybenzyl group—also called a vanillyl group—circled in red, (**B**) capsaicin (PubChem CID 1548943), (**C**) 6-gingerol (PubChem CID 442793), (**D**) ferulic acid (PubChem CID 445858), and (**E**) incarvillateine (PubChem CID 9875096).

**Figure 2 pharmaceuticals-16-00917-f002:**
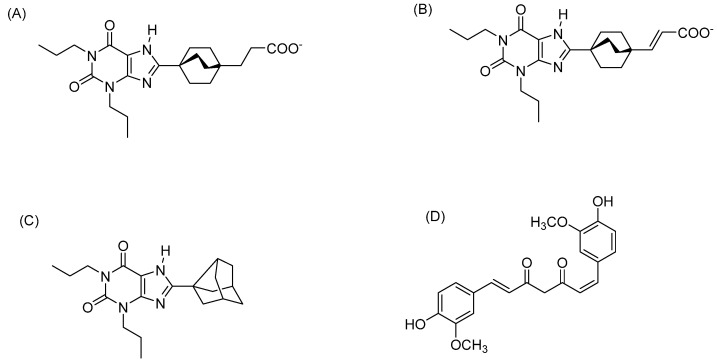
Chemical structures of (**A**) compound A, (**B**) compound B, and (**C**) KW3902 (NAX), used as the control for the docking analysis, and (**D**) CTCUR in its EZ-keto form.

**Figure 3 pharmaceuticals-16-00917-f003:**
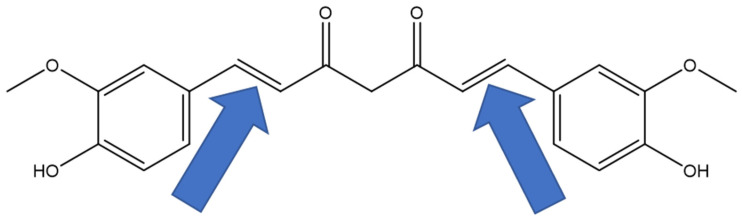
Chemical structure of curcumin (PubChem CID 969516) with the two non-aromatic, carbon–carbon double bonds highlighted with blue arrows. The isomerization process used in this study altered one of these double bonds, leaving the other unchanged.

**Figure 4 pharmaceuticals-16-00917-f004:**
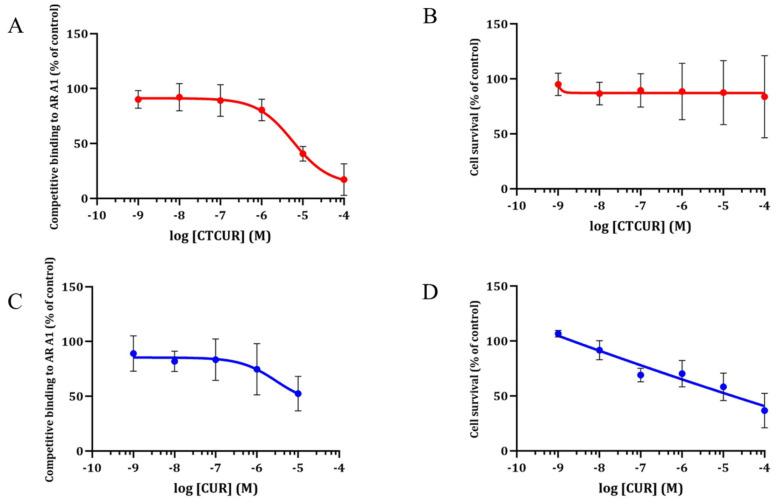
(**A**,**C**) Binding curves at A_1_AR showing the competition of a fluorescent AR ligand (CA200623) with (**A**) CTCUR (*n* = 3, K_i_ 306 nM) and (**C**) CUR (*n* = 3, K_i_ = not measurable). (**B**,**D**) The survival curves for the A_1_AR-transfected cells were treated with (**B**) CTCUR (*n* = 5) and (**D**) CUR (*n* = 3). Red lines refer to treatments with CTCUR and blue lines are treatments with CUR. The data represent the mean ± the SEM with respect to control with no treatment. Each treatment had three to five replicates.

**Figure 5 pharmaceuticals-16-00917-f005:**
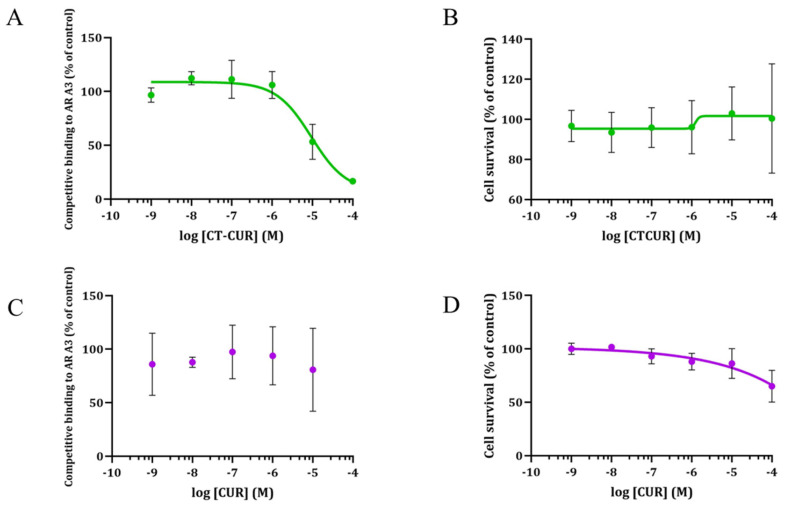
(**A**,**C**) Binding curves at A_3_AR showing the competition of a fluorescent AR ligand (CA200623) with (**A**) CTCUR (*n* = 3, K_i_ = 400 nM) and (**C**) CUR (*n* = 3, K_i_ > 10,000,000 nM). (**B**,**D**) The survival curves for the A_3_AR-transfected cells were treated with (**B**) CTCUR (*n* = 3) and (**D**) CUR (*n* = 3). Green lines refer to treatments with CTCUR and purple dots/lines are treatments with CUR. The data represent the mean ± the SEM with respect to control without any treatment. Each treatment had three to five replicates.

**Figure 6 pharmaceuticals-16-00917-f006:**
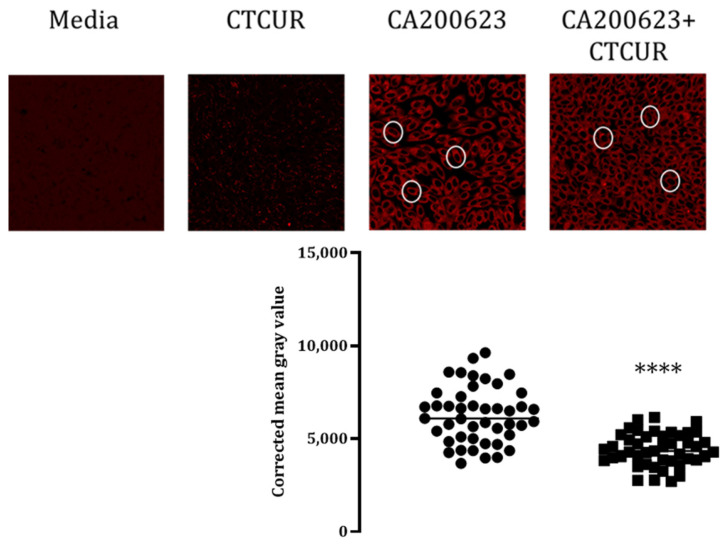
Competitive binding assay of CTCUR and a fluorescent AR ligand (CA200623) at A_1_AR, viewed through a confocal microscope. The white circles highlight individual cells. For the CA200623 treatment, two wells were selected, three image stacks were collected from each well, and nine cells were selected from each image stack. The same process was conducted for the CA200623 + CTCUR treatment. One stack from each treatment was discarded due to low image quality. Therefore, the total number of cells imaged was 90. Each dot within the corrected mean gray value scatterplot represents one cell. Each cluster of dots corresponds to the treatment indicated above it (**** indicates *p* < 0.0001, *n* = 3).

**Figure 7 pharmaceuticals-16-00917-f007:**
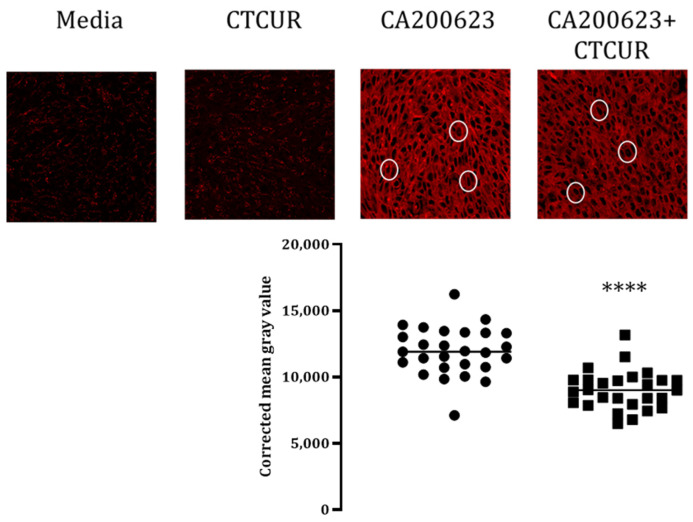
Competitive binding assay of CTCUR at A_3_AR viewed through a confocal microscope. The white circles highlight individual cells. For the CA200623 treatment, two wells were selected, three image stacks were collected from each well, and nine cells were selected from each image stack. The same process was conducted for the CA200623 + CTCUR treatment. Three stacks from each treatment were discarded due to low image quality. Therefore, the total number of cells imaged was 54. Each dot within the corrected mean gray value scatterplot represents one cell. Each cluster of dots corresponds to the treatment indicated above it (**** indicates *p* < 0.0001, *n* = 3).

**Figure 8 pharmaceuticals-16-00917-f008:**
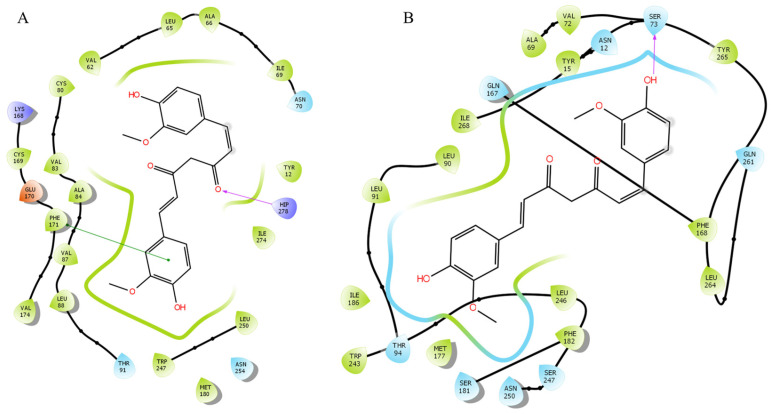
Docking analysis of the AR/CTCUR complex. (**A**) Interactions with A_1_AR. ΔG = −10.22 kcal/mol; (**B**) interactions with A_3_AR, ΔG = −10.16 kcal/mol. The H-bonds were presented in magenta arrows and aromatic-aromatic interactions in green lines.

**Table 1 pharmaceuticals-16-00917-t001:** Binding affinities of CUR and CTCUR in the context of other important AR ligands. The values are K_i_ (nM) at the human versions of each AR subtype.

Compound *	A_1_	A_3_	A_2A_	A_2B_
CTCUR	306	400	5107 ^a^	6722 ^a^
CUR	-	>10,000,000	>10,000,000 ^a^	5,700,000 ^a^
Adenosine	100 ^b^	290 ^b^	310 ^b^	15,000 ^b^
NECA	14 ^c^	25 ^c^	20 ^c^	140 ^c^
Tecadenoson	2 ^d^	227 ^d^	6390 ^d^	25,800 ^d^
IB-MECA	51 ^c^	1.8 ^c^	2900 ^c^	11,000 ^c^
Caffeine	10,700 ^c^	13,300 ^c^	23,400 ^c^	33,800 ^c^
Theophylline	6770 ^c^	22,300 ^c^	1710 ^c^	9070 ^c^

^a^ [38], ^b^ [39], ^c^ [20], ^d^ [40]; * adenosine is an endogenous agonist and prescription drug. NECA is an agonist used in AR research. Tecadenoson is an A_1_AR-selective agonist [41]. IB-MECA (also known as piclidenoson) is an A_3_AR-selective agonist [42], caffeine is an antagonist, and theophylline is a non-selective antagonist used in AR research.

**Table 2 pharmaceuticals-16-00917-t002:** Summary of the results from the three methods used to measure the binding affinity of CTCUR with the four AR subtypes. The A_2A_ and A_2B_ data were from a previous study (Hamilton et al., 2021). The data suggested that CTCUR was selected for the G_i_-linked ARs over the G_s_-linked ARs.

Method	A_1_	A_3_	A_2A_	A_2B_
CBA (K_i in_ nM)	306	400	5107	6722
Confocal Microscopy	Confirms binding (*p* < 0.0001)	Confirms binding (*p* < 0.0001)	No difference (N/A)	Confirms binding (*p* < 0.05)
Docking				
(ΔG in kcal/mol)	−10.22	−10.16	−9.6	−7.3

**Table 3 pharmaceuticals-16-00917-t003:** Docking scores (kcal/mol), experimental free energy (kcal/mol), and relative errors (ΔΔG, kcal/mol) between ΔGpred (binding score) and ΔGexp for the A_1_AR and A_3_AR subtypes.

Compounds	Docking (A_1_)	Docking (A_3_)	ΔGexp ^a^ (A_1_)	ΔGexp ^a^ (A_3_)	ΔΔG (A_1_)	ΔΔG (A_3_)
CTCUR	−10.22	−10.16	−8.89	−8.73	−1.33	−1.43
Tecadenoson	−7.00	−9.29	−11.87	−9.06	4.87	−0.23
Adenosin	−7.60	−8.08	−9.55	−8.92	1.95	0.84
Theophylline	−7.42	−7.68	−7.04	−6.35	−0.38	−0.63
Caffeine	−5.77	−6.73	−5.93	−6.65	0.16	−0.08
Vanillin	−6.39	−7.30				
Vapsaicin	−7.19	−8.53				
6-gingerol	−8.68	−9.33				
Ferulic acid	−6.64	−7.55				

^a^: ΔG_exp = RTlnK_i_ = 1.987 ∗ 298.15 ∗ ln((Ki in nM) × 10^−9^)/1000 using the K_i_ values reported in Table 1.

## Data Availability

Data is contained within the article and Appendix A.

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
