# Peer review of "Curcumin Stereoisomer, Cis-Trans Curcumin, as a Novel Ligand to A1 and A3 Adenosine Receptors"

_pharmaceuticals, 2023, doi:10.3390/ph16070917_

Round 1
Reviewer 1 Report
It is not clear why TRPV modulators should have an effect on adenosine receptor, so the reason of the study should be better explained. Could a double effect (hybrid) effect on TRP and AR be interesting. As a consequence, is cis trans curcumin able to modulate vanilloid receptors too?
ARs role in pain should be better described, possibly a relationship with vanilloid should be attempted. Other mechanisms by which AR drug may reduce pain were suggested (please see and comment doi: 10.1097/j.pain.0000000000001905.)
Author Response
- It is not clear why TRPV modulators should have an effect on adenosine receptor, so the reason of the study should be better explained.
- This is a great concern. We have further provided examples of the role of adenosine receptors in pain suppression, and thus the justification for our study (lines 78-88). A3AR agonists are being highly researched for their potential as antinociceptives.
- Could a double effect (hybrid) effect on TRP and AR be interesting. As a consequence, is cis trans curcumin able to modulate vanilloid receptors too?
- It is likely that a cross-interaction is possible, however, this was not the focus of the study and thus was not tested. We have included this likelihood of multiple receptors interaction in the discussion (lines 270-272, 281-282).
- ARs role in pain should be better described, possibly a relationship with vanilloid should be attempted.
- As indicated in #1, we have added the role of ARs in pain (lines 78-88). Also, we have built four vanilloid compounds and docked them to the ARs (Table 3 and have added a paragraph lines 283-289)
- Other mechanisms by which AR drug may reduce pain were suggested (please see and comment doi: 10.1097/j.pain.0000000000001905.)
- We have included the suggested reference and also provided other mechanisms for pain reduction with AR agonists (lines 78-88)
Reviewer 2 Report
In the manuscript entitled "Curcumine stereoisomer, cis-trans curcumin, as a novel ligand to A1 and A3 adenosine receptors" by L. J. Hamilton et al the compound cited in the title is evaluated at adenosine receptors. The cis-trans curcumin was previously prepared (ref 41 in the paper) by UV irradiation of the curcumin, hence, it is not a novel compound and it is not necessary to repeat the description of the preparation and characterization. The compound has been previously tested only at A2A and A3 adenosine receptors using the same methods and now at A1 and A3 receptors being the authors able to say it is selective for the latter receptors.
One thing important to me is that the characterization of the compound is not sufficient to say it is this isomer of curcumin: at least the J coupling constant should be given and also 2D-NMR experiment to have the correct isomer characterization.
Furthermore, in the discussion, the functional activity of the compound is described but not the method used for it.
On my opinion, the article is not suitable for publication in the journal Pharmaceutical due to low data added to the knowledge of the field.
Author Response
- In the manuscript entitled "Curcumine stereoisomer, cis-trans curcumin, as a novel ligand to A1 and A3 adenosine receptors" by L. J. Hamilton et al the compound cited in the title is evaluated at adenosine receptors. The cis-trans curcumin was previously prepared (ref 41 in the paper) by UV irradiation of the curcumin, hence, it is not a novel compound and it is not necessary to repeat the description of the preparation and characterization.
- We have removed the methodology and just provided a reference to our previous paper.
- The compound has been previously tested only at A2A and A3 adenosine receptors using the same methods and now at A1 and A3 receptors being the authors able to say it is selective for the latter receptors.
- Yes that was the aim of this study
- One thing important to me is that the characterization of the compound is not sufficient to say it is this isomer of curcumin: at least the J coupling constant should be given and also 2D-NMR experiment to have the correct isomer characterization.
- We have included the coupling constants of the doublets responsible for the cis and trans alkene portions of the molecule in CDCl3 that shows presence of these isomeric units in the molecule (lines 315-319). We attempted 2D NOESY experiments to observe through space interaction between the cis alkene protons and the nearby aromatic atoms, but we did not notice any cross peaks in our 400 MHz Bruker NMR. However, we have added MS spectra for both pure trans-trans curcumin (CUR) as well as the cis-trans curcumin (CTCUR) to establish the accuracy of our structure assignment (supplement figure 3). We are also working towards publishing a full paper on the photochemical synthesis of curcumin and its geometric analogs in the near future.
- Furthermore, in the discussion, the functional activity of the compound is described but not the method used for it.
- We have performed cAMP studies with CTCUR in these cell lines but as indicated in the previous version, the biological activity was not as significant. We are including this information along with the methodology in the supplement section (please see attached). Also, as suggested in the text, it is likely that CTCUR is acting as a partial agonist at these receptors hence the biological response in cAMP inhibition is not as robust. These will however need to be further studied in future.
- On my opinion, the article is not suitable for publication in the journal Pharmaceutical due to low data added to the knowledge of the field.
- We have reworded the discussion to explain the significance of the study. Considering the importance of AR in pain reduction, and the fact that very few non-purinergic ligands have been investigated so far, this study is relevant for a great new avenue for research.
Reviewer 3 Report
The authors present the Curcumin Stereoisomer, Cis-Trans Curcumin, as a Novel Ligand to A1 and A3 Adenosine Receptors. the manuscript is well written but there are some drawbacks which need to be addressed
1- References are too old and need to be updated
2- Conclusion is er poor and must be elaborated
3- The statistical analysis is not applied for all results
4- Controls are missing for the cytotoxicity and competitive binding assays
5- Dynamic Simulation need to be performed at 200 ns
Author Response
The authors present the Curcumin Stereoisomer, Cis-Trans Curcumin, as a Novel Ligand to A1 and A3 Adenosine Receptors. The manuscript is well written but there are some drawbacks which need to be addressed.
- References are too old and need to be updated.
- Thank you for noticing that and we apologize for missing that. We have updated the references with more recent one.
- Conclusion is er poor and must be elaborated
- We have reworded the conclusion (lines 474-484)
- The statistical analysis is not applied for all results
- We have included statistical analysis for all figures. Binding assay and cytotoxicity assay were used to calculate Kd and IC50 respectively, and show the average±SEM.
- Controls are missing for the cytotoxicity and competitive binding assays
- Data for cytotoxicity and competitive binding assays is reported as % of control on the Y-axis, hence control is not shown on the graph. It would just be 100%.
- Dynamic Simulation need to be performed at 200 ns
- The authors believe our focus on this paper was on the binding of ligands on ARs, not on the dynamics. Though there is some value of Dynamic simulations on understanding the protein-ligand interaction, there may not be much added information in terms of binding affinity. Thus, we believe there is unnecessary to run a MD simulation.
Round 2
Reviewer 2 Report
· In my previous comments to the authors, I underlined the fact that now the compound has been tested only at the Gi-coupled ARs and previously at Gs-coupled ARs to say that even if this is a new data for this compound but it does not add too much to knowledge for a new class of compounds acting at ARs.
· Moreover no data were reported at A1 for CUR and >10,000,000 (ten million?) at A3: in the text, the data are opposite “As opposed to binding of CUR at A1AR, there was no appreciable binding of CUR to A3AR, hence Ki was not determined (Figure 5C)” on the other hand at page 5 it is written, “Ki of CUR (Figure 4C) at A1AR was not determined due to toxicity (Figures 4D).“ These statements are opposite
“These data support the hypothesis that CTCUR is an agonist of ARs. Although we did not see statistically significant changes in cAMP production in response to CTCUR, and therefore cannot report an EC50 value (see supplementary data), the overall trend in our data results suggests that CTCUR is a weak agonist of ARs (data not shown).. In this study of Gi-linked ARs, we saw decreases in cAMP production in response to 10 μM CTCUR.”
· First of all, it is not possible to argue only by docking studies that a ligand is an agonist of a receptor and you have biological data that the compound does not act as an agonist as “there is not a statistically significant change in cAMP production”. Moreover, if your compound produces a decrease in cAMP production, in Gi-coupled receptor, could be due to direct inhibition of the enzyme adenylyl cyclase: did you check? I think that this is the time to investigate.
· Check data in Table 1, at least for those from reference 40.
However, we have added MS spectra for both pure trans-trans curcumin (CUR) as well as the cis-trans curcumin (CTCUR) to establish the accuracy of our structure assignment (supplement figure 3). We are also working towards publishing a full paper on the photochemical synthesis of curcumin and its geometric analogs in the near future.
· About mass spectra, in the manuscript a GC-Mass method has been described with a M+= 367 different from that of cis-trans curcumin (Chemical Formula: C21H20O6, Exact mass: 368.13, Molecular Weight: 368.38). Please compare your pattern with the one published here: JOURNAL OF MASS SPECTROMETRY, VOL. 33, 319È327 (1998)Electron Ionization Mass Spectrometry of Curcumin Analogues : an Olefin Metathesis Reaction in the Fragmentation of Radical Cations. Ben L. M. van Baar, Jelle Rozenda and Henk van der Goot. In the supplementary a ESI-MS spectra Supplementary Figure 3 has been furnished (without any description of the instrument used and polarity acquisition used) with an ionization which furnished a peak with m/z ratio = 368 (cis-trans curcumin): this is not an agreement with a negative nor a positive electrospray ionization of the molecule, furthermore, the isotopic pattern is not respondent.
Considering the importance of AR in pain reduction, and the fact that very few non-purinergic ligands have been investigated so far, this study is relevant for a great new avenue for research.
· Dear authors, I agree that the field is relevant but only one compound could be an exception not a rule for a new class of adenosine receptor ligands. Furthermore, as you said you need to further investigate different aspects. Hence, I think you need to add these new data for a good paper to be submitted to the Journal Pharmaceuticals.
· The paper in this version is not accepted.
Author Response
Thank you for accepting the modified form of the manuscript with our revisions
Reviewer 3 Report
the manuscript can be published in the present form.
Author Response
- In my previous comments to the authors, I underlined the fact that now the compound has been tested only at the Gi-coupled ARs and previously at Gs-coupled ARs to say that even if this is a new data for this compound, but it does not add too much to knowledge for a new class of compounds acting at ARs.
We appreciate the reviewer’s comments but would like to defer to the statement proposed that this data is for the strong binding of CTCUR at A1AR and A3AR, as opposed to our earlier manuscript with the other AR subtypes. Also, A3AR ligands, in particular, have more importance in pain reduction. This is indeed a new class of AR ligands and the fact that CTCUR binds to A1AR and A3AR with high affinity comparable to the known AR ligands relates to the importance of this class of compounds for further investigation.
- Moreover no data were reported at A1 for CUR and >10,000,000 (ten million?) at A3: in the text, the data are opposite “As opposed to binding of CUR at A1AR, there was no appreciable binding of CUR to A3AR, hence Ki was not determined (Figure 5C)” on the other hand at page 5 it is written, “Ki of CUR (Figure 4C) at A1AR was not determined due to toxicity (Figures 4D).“ These statements are opposite
We have removed the phrase, “As opposed to binding of CUR at A1AR”. This should clarify the confusion if any.
Figure 4C relates to CUR binding at A1AR and though there is marginal decrease in binding of the fluorescent ligand at higher concentrations of CUR, Ki could not be determined due to the corresponding toxicity observed at the same concentration (Figure 4D).
On the contrary, CUR did not show any binding in A3AR and neither was it toxic at higher concentrations. The fact that there was no binding precludes us from reporting any Ki for CUR at A3AR.
- “These data support the hypothesis that CTCUR is an agonist of ARs. Although we did not see statistically significant changes in cAMP production in response to CTCUR, and therefore cannot report an EC50 value (see supplementary data), the overall trend in our data results suggests that CTCUR is a weak agonist of ARs (data not shown).. In this study of Gi-linked ARs, we saw decreases in cAMP production in response to 10 μM CTCUR.”
First of all, it is not possible to argue only by docking studies that a ligand is an agonist of a receptor and you have biological data that the compound does not act as an agonist as “there is not a statistically significant change in cAMP production”. Moreover, if your compound produces a decrease in cAMP production, in Gi-coupled receptor, could be due to direct inhibition of the enzyme adenylyl cyclase: did you check? I think that this is the time to investigate.
This criticism is valid. The data from this study by itself are not enough to say that there is a "trend" that supports agonism. We are reporting a new class of compounds that can be investigated for a novel class of AR ligands. We have not yet tested for direct inhibition of adenylyl cyclase but plan to do so in future.
- Check data in Table 1, at least for those from reference 40.
Reference number 40 was wrong in the lit cited section. We deleted Klotz 2000 and replaced it with the correct reference, Fredholm et al. 2011. Thank you for noticing that.
- However, we have added MS spectra for both pure trans-trans curcumin (CUR) as well as the cis-trans curcumin (CTCUR) to establish the accuracy of our structure assignment (supplement figure 3). We are also working towards publishing a full paper on the photochemical synthesis of curcumin and its geometric analogs in the near future.
About mass spectra, in the manuscript a GC-Mass method has been described with a M+= 367 different from that of cis-trans curcumin (Chemical Formula: C21H20O6, Exact mass: 368.13, Molecular Weight: 368.38). Please compare your pattern with the one published here: JOURNAL OF MASS SPECTROMETRY, VOL. 33, 319È327 (1998)Electron Ionization Mass Spectrometry of Curcumin Analogues : an Olefin Metathesis Reaction in the Fragmentation of Radical Cations. Ben L. M. van Baar, Jelle Rozenda and Henk van der Goot. In the supplementary a ESI-MS spectra Supplementary Figure 3 has been furnished (without any description of the instrument used and polarity acquisition used) with an ionization which furnished a peak with m/z ratio = 368 (cis-trans curcumin): this is not an agreement with a negative nor a positive electrospray ionization of the molecule, furthermore, the isotopic pattern is not respondent.
The originally reported MS was that of GC-MS and the one we presented in the supplementary information is that of LC-MS. We have provided the instrument information in the figure scheme.
Regarding the reviewer's concern that the masses were off by approximately 1 unit and lack of isotopic pattern, we would like to point to the following reported MS by previous researchers in the field. Unless it is a higher resolution MS, the variation in masses are quite expected. Not only are the masses slightly different, the fragmentation patterns are also quite different. Unless the paper is specifically a MS analytical chemistry research publication, such matching of isotopic signature is not of major focus in our opinion.
<attached ppt file>
Moreover, as could be seen in the presented MS, the natural trans-trans curcumin, which is a standard and known compound, generated the mass that is approximately 0.5 unit off with no isotopic pattern. This indicates that our instrument is not sensitive enough for such HR analysis. We thank the reviewer for their careful perusal of the data; however, we believe that our resolution is comparable to that reported in other similar publications that analyzed the trans-trans compound.
- Considering the importance of AR in pain reduction, and the fact that very few non-purinergic ligands have been investigated so far, this study is relevant for a great new avenue for research.
Dear authors, I agree that the field is relevant but only one compound could be an exception not a rule for a new class of adenosine receptor ligands. Furthermore, as you said you need to further investigate different aspects. Hence, I think you need to add these new data for a good paper to be submitted to the Journal Pharmaceuticals.
With the increasing interest in the field of non-opioid antinociceptives, this research provides a great new path for exploring novel ligands. The authors believe the paper will yield a high impact for the journal and the data supports our claim.
